# Transcriptional Analysis of lncRNA and Target Genes Induced by Influenza A Virus Infection in MDCK Cells

**DOI:** 10.3390/vaccines11101593

**Published:** 2023-10-14

**Authors:** Geng Liu, Mengyuan Pei, Siya Wang, Zhenyu Qiu, Xiaoyun Li, Hua Ma, Yumei Ma, Jiamin Wang, Zilin Qiao, Zhongren Ma, Zhenbin Liu

**Affiliations:** 1Engineering Research Center of Key Technology and Industrialization of Cell-Based Vaccine, Ministry of Education, Lanzhou 730030, China; lg13619271425@163.com (G.L.); peimengyuan1998@163.com (M.P.); wsy19991216@163.com (S.W.); qiuzhenyu9810@163.com (Z.Q.); lixy000920@163.com (X.L.); jiaminwang1987@163.com (J.W.); qiaozilin@xbmu.edu.cn (Z.Q.); mazhr@foxmail.com (Z.M.); 2Gansu Tech Innovation Center of Animal Cell, Biomedical Research Center, Northwest Minzu University, Lanzhou 730030, China; 3Key Laboratory of Biotechnology and Bioengineering of National Ethnic Affairs Commission, Biomedical Research Center, Northwest Minzu University, Lanzhou 730030, China; 4Gansu Provincial Bioengineering Materials Engineering Research Center, Lanzhou 730010, China; 13919410993@163.com (H.M.); mayumei16@163.com (Y.M.)

**Keywords:** IAV, MDCK, lncRNAs, transcriptomics, influenza vaccine

## Abstract

Background: The MDCK cell line is the primary cell line used for influenza vaccine production. Using genetic engineering technology to change the expression and activity of genes that regulate virus proliferation to obtain high-yield vaccine cell lines has attracted increasing attention. A comprehensive understanding of the key genes, targets, and molecular mechanisms of viral regulation in cells is critical to achieving this goal, yet the post-transcriptional regulation mechanism involved in virus proliferation—particularly the effect of lncRNA on influenza virus proliferation—is still poorly understood. Therefore, this study used high-throughput RNA-seq technology to identify H1N1 infection-induced lncRNA and mRNA expression changes in MDCK cells and explore the regulatory relationship between these crucial lncRNAs and their target genes. Results: In response to H1N1 infection in MDCK cells 16 h post-infection (hpi) relative to uninfected controls, we used multiple gene function annotation databases and initially identified 31,501 significantly differentially expressed (DE) genes and 39,920 DE lncRNAs (|log2FC| > 1, *p* < 0.05). Among these, 102 lncRNAs and 577 mRNAs exhibited predicted correlations with viral response mechanisms. Based on the magnitude of significant expression differences, related research, and RT-qPCR expression validation at the transcriptional level, we further focused on 18 DE mRNAs and 32 DE lncRNAs. Among these, the differential expression of the genes RSAD2, CLDN1, HCLS1, and IFIT5 in response to influenza virus infection was further verified at the protein level using Western blot technology, which showed results consistent with the RNA-seq and RT-qPCR findings. We then developed a potential molecular regulatory network between these four genes and their six predicted lncRNAs. Conclusions: The results of this study will contribute to a more comprehensive understanding of the molecular mechanism of host cell non-coding RNA-mediated regulation of influenza virus replication. These results may also identify methods for screening target genes in the development of genetically engineered cell lines capable of high-yield artificial vaccine production.

## 1. Background

The annual influenza epidemic poses a major threat to public health and causes a large number of deaths, especially among the elderly aged 65 and above [1]. Influenza vaccination is an effective means to prevent and control influenza outbreaks [2]. Compared with the traditional chicken embryo-based vaccine production process, the mammalian cell-based process can produce a larger vaccine yield more quickly, which is suitable for an industrial production scale. The use of closed reactors in the production process can prevent bacterial contamination and reduce the chance of hemagglutinin mutations in the production process, thereby improving the effectiveness of the vaccine and allowing for a quick response to an influenza pandemic [3]. Therefore, the cell matrix-based method for inactivated viral vaccine production has gradually become mainstream [4]. The MDCK cell line is the primary cell line for influenza vaccine production due to its high virus infection efficiency, rapid cell proliferation, and resistance to mutation. It is widely used in influenza virus-related basic research and vaccine production [5]. The use of gene editing in cells to inhibit the antiviral signaling pathway and promote virus replication can effectively increase both the cellular viral load and vaccine yield and reduce production costs [6]. The core problem is to fully understand the key genes and their regulatory mechanisms that affect influenza virus replication in MDCK cells. 

Scientists have increased their understanding of the interaction mechanism between host cells and influenza viruses, with certain key genes reported to play a crucial role [7]. For example, when infected with the influenza A virus (IAV), host cells can recognize viral pathogen-associated molecular patterns (PAMPs) using various host pattern recognition receptors (PRRs) [8]. These PRRs, which include the Toll-like receptor (TLR), RIG-I-like receptor (RLR), and NOD-like receptor (NLR), play a role in the antiviral response by recognizing the IAV nucleic acid and proteins and, in turn, activating a series of complex signaling pathways, inducing the production of interferon (IFN) and inflammatory factors, and initiating the expression of downstream antiviral IFN-stimulated genes (ISGs) [9]. Currently, the signaling pathways known to be involved in the antiviral response include the mitogen-activated protein kinases (MAPKs), the nuclear factor (NF)-κB, the PKC/PKR, the PI3K/AKT, and the JAK-STAT signaling pathways [10]. Activation of the TLR family member TLR2 and its related signaling pathways has been shown to induce the innate immune response, which can then resist various respiratory virus infections, including influenza and the globally dominant SARS-CoV-2 viruses [11]. For example, the SARS-CoV-2 envelope (E) protein has been shown to specifically interact with the TLR2 receptor, which in turn plays its antiviral role by activating the NF-κB transcription factor and stimulating the production of the inflammatory chemokine CXCL8 [12]. In addition, some studies have shown that TLR2, IFIT5, RSAD2, CLDN1, and HCLS1 play an important role in regulating the replication of influenza and other viruses in host cells [13,14,15,16,17,18,19].

Long non-coding RNAs (lncRNAs) are RNAs greater than 200 nt in length; although they lack the ability to encode proteins, they serve critical functions [20]. lncRNAs can combine with DNA, RNA, and proteins to regulate gene expression at multiple levels, including epigenetic, transcriptional, and post-transcriptional regulation. Studies have shown that lncRNAs play a key regulatory role in the fight between host cells and viruses. Recently, they have been shown to interact with a variety of PRRs and both directly or indirectly act on their target genes or related signaling pathways to regulate IAV replication [21]. For example, lnc-Lsm3b inhibits the initiation of antiviral signal transduction downstream of RIG-I by competing with viral RNA for the binding of RIG-I monomers, thus promoting the replication of A(H1N1) [22]. Additionally, lnc-MxA is an ISG that forms an RNA-DNA triplet, which inhibits IFN-β and subsequent activation of its related signaling pathway, thereby promoting IAV replication [23]. However, even with the current level of knowledge regarding host cell regulation of influenza virus proliferation, there are still few reports of the effect of lncRNAs specifically on influenza virus replication. Therefore, these lncRNA functions require further exploration, especially in MDCK cells. In this study, RNA-seq transcriptomics was used to explore differentially expressed (DE) lncRNAs and mRNAs in MDCK cells infected with A(H1N1). Real-time PCR (qPCR) and Western blotting techniques were used to verify the expression of screened genes. Furthermore, databases were combed to predict the regulatory network between lncRNAs and targeted mRNAs that may participate in the regulation of influenza virus proliferation in MDCK cells. These findings will contribute to a more comprehensive understanding of the molecular mechanism underlying the post-transcriptional regulation of influenza virus replication in host cells. These methods can also provide a basis for screening target genes to produce genetically engineered cell lines capable of high-yield vaccine production.

## 2. Results

### 2.1. Characterization of IAV Infection in MDCKs

Infection with influenza A virus for 16 h is an early activation time point for innate immune response in MDCK host cells. Moreover, we conducted screening experiments on MDCK cells at different time points (12 hpi, 16 hpi, 24 hpi, 36 hpi) infected with the X-275 strain (H1N1) virus with an MOI value of 0.01. As shown in Figure 1, at 16 hpi, it was found that the expression levels of A(H1N1) viral NP and NS1 genes were the highest. Therefore, 16 hpi was chosen as the condition for preparing cell samples for high-throughput RNA sequencing (RNA seq).

### 2.2. Overview of Sequencing Data

We identified the DEGs between MDCK cells infected with influenza virus H1N1 (X-275 strain) 16 hpi (MDCK + H1N1) relative to uninfected MDCK cells (MDCK) using |log2FC| > 1 and *p* < 0.05 as the screening criteria. The volcano plot results show that 31,501 total mRNAs were significantly differentially expressed in MDCK cells infected with H1N1 relative to uninfected cells, of which 15,439 were up-regulated, and 16,062 were down-regulated (Figure 2A, Appendix A). Additionally, 39,920 total lncRNAs were significantly differentially expressed in infected MDCK cells relative to controls, of which 38,002 were up-regulated and 1918 were down-regulated (Figure 2B, Appendix A). Furthermore, we used heat maps to display the DE mRNAs (Figure 2C) and lncRNAs (Figure 2D), which showed that the mRNA and lncRNA expression pattern between influenza virus H1N1-infected and uninfected MDCK cells was significantly different and that the expression pattern among samples within the same treatment group was relatively stable with good repeatability.

### 2.3. Screening of Virus-Related DEGs Revealed Enrichment Changes in Several Signaling Pathways in MDCK Cells after Infection

In the preliminary screening of biological processes related to viral immune response using the GO database, we screened 577 DE mRNAs related to virus response, of which 280 were significantly up-regulated, and 297 were down-regulated in infected MDCK cells relative to uninfected cells (Appendix A). To screen the genes involved in the MDCK virus response and match them with predicted lncRNAs, we used these 577 preliminarily screened virus response-related DE mRNAs to perform intersection analysis with the predicted target genes of the lncRNAs from the sequencing results. From this analysis, we identified 141 virus response-related DE mRNAs matching predicted DE lncRNAs (Figure 3A), of which 88 were up-regulated and 53 were down-regulated (Appendix A). We further analyzed the function, distribution, and signaling pathways of these preliminarily screened DEGs using GO enrichment and KEGG pathway cluster analysis. The GO enrichment analysis showed that the BP results of the 577 screened DE mRNAs were all related to the immune response and the host cell response to virus, such as the antiviral response (GO:0051607, *p* < 0.05), I-κB kinase/NF-κB signal transduction (GO:007249, *p* < 0.05), innate immune response (GO:0045087, *p* < 0.05), interleukin-1 mediated signaling pathway (GO:0070498, *p* < 0.05), Type I interferon signaling pathway (GO:0060337, *p* < 0.05), virus entry into host cells (GO:0046718, *p* < 0.05), and negative regulation of viral genome replication (GO:0045071, *p* < 0.05). In addition, the CC and MF results of the screened DE mRNAs were also related to viral infection and the immune response, such as viral receptor activity (GO:0001618, *p* < 0.05) and NF-κB binding (GO:0051059, *p* < 0.05) (Figure 3B). The KEGG pathway enrichment analysis of our 577 screened DE mRNAs was consistent with the analysis results from our total DE mRNAs, primarily revealing enrichment in pathways such as influenza A infection (*p* < 0.05), the NF-κB signaling pathway (*p* < 0.05), the TNF signaling pathway (*p* < 0.05), and HTLV-I infection (*p* < 0.05) (Figure 3C); these are classical signaling pathways related to inflammation and viral infection that are enriched with a substantial number of DEGs. Although further experimental verification is required to determine whether the DEGs in these signaling pathways are the key factors affecting IAV infection in MDCK cells, they provide potential targets for our subsequent research. 

### 2.4. Verification of mRNA Expression of 18 Screened DEGs

We preliminarily screened 141 differentially expressed mRNA based on transcriptome analysis results. On this basis, we selected the top 20 genes with the highest differences in both up-regulation and down-regulation based on the multiple and significance of expression differences, totaling 40 genes. Considering the possibility of some randomness and false positive results in high-throughput transcriptome analysis results. We used real-time PCR to perform three verifications on samples of different cell generations and screened a total of 18 mRNA with uniform expression differences and consistent transcriptome analysis results. The results showed that 6 of these mRNAs were up-regulated (HPX, RHOH, TLR2, HCLS1, IFIT5, and RSAD2) and 12 of these mRNAs were down-regulated (AHNAK, CLDN1, SQSTM1 (P62), MAP3K7 (Tak1), AP2B1, ERC1, EXOSC4, HNRNPR, IRF2, LRSAM1, NF2, and TKFC (TKFCD)) in MDCK + H1N1 cells relative to uninfected MDCK cells (Figure 4). Furthermore, these differential expression patterns were consistent with the RNA-seq results. Five of our screened DEGS have been reported to affect influenza virus replication in host cells: TLR2, IFIT5, and RSAD2 have been reported to inhibit replication, while HCLS1 and AP2B1 have been shown to promote replication [13,14,15,16,17]. The remaining 13 screened DEGs have also been shown to play an important role in a variety of viruses. For example, CLDN1 promotes the replication of the hepatitis C virus in host cells via the PI3K-AKT pathway, and IFIT5 promotes the replication of EV-HPV in host cells via the TCR signaling pathway [18,19]. The four DEGs RSAD2, HCLS1, IFIT5, and CLDN1 exhibited substantially high relative mRNA expression. Therefore, these four genes were selected for subsequent protein expression verification experiments.

### 2.5. Prediction and Expression Verification of lncRNA

Since the GO and KEGG pathways analyses indicated that the 18 screened DE mRNAs had a potential regulatory relationship and interaction in response to IAV infection, we then used omics analysis to screen 37 lncRNAs potentially matching these 18 mRNAs. Among them, 35 lncRNAs have a positive regulatory effect on their target gene mRNAs, and two lncRNAs have a negative regulatory effect on their target gene mRNAs; furthermore, our results predicted that 36 lncRNAs regulate their target gene mRNAs via cis action, while one lncRNA regulates its target gene via trans action (Appendix A). We then used RT-qPCR to verify the relative expression of these lncRNAs. Our expression validation analysis showed 32 lncRNAs with a consistent expression between the RT-qPCR and omics results, with 16 potential target gene mRNAs matching these 32 lncRNAs (Figure 5). Among them, 30 lncRNAs have positive regulatory effects on their target gene mRNAs, and two lncRNAs have negative regulatory effects on their target gene mRNAs. Additionally, 31 lncRNAs regulate their target gene mRNAs through cis action, and one lncRNA regulates its target gene through trans action (Table 1). Furthermore, our analysis identified 30 lncRNAs for the first time, while the remaining two lncRNAs sequences were previously known. The 30 newly predicted lncRNA sequences are shown in Appendix A. The mechanism of cis-regulation is very complex, involving numerous regulatory effects on the polymerase and promoter, which we will explore in further research [24]. 

### 2.6. Verification of Protein Expression of Four Target DEGs

Because the regulation of some lncRNAs on their target genes is reflected in changes at the post-transcriptional level, we further verified the changes in expression at the protein level using Western blot. Among the 18 genes that were screened at the mRNA level, we further selected four genes to test at the protein level based collectively on existing gene function reports and the magnitude of RT-qPCR expression differences. Western blot results showed that the protein expression levels of RSAD2, HCLS1, and IFIT5 increased and CLDN1 decreased in response to H1N1 infection (Figure 6, Full-length blots/gels are presented in Appendix A); moreover, this protein expression trend was consistent with the mRNA expression trend. Among these four proteins, RSAD2 was highly expressed at both the mRNA and protein levels, as well as in the RNA-seq results; therefore, our subsequent research will focus on this gene. 

### 2.7. Analysis of the lncRNA-mRNA Interaction Network Potentially Affecting IAV Replication in MDCK Cells

By combining our previous screening results for 18 DEGs and 32 matched DE lncRNAs with predicted histone expression trends, RT-qPCR and Western blot expression verification, the pre-test regulation mode of the 32 lncRNAs from RNA-seq (i.e., either cis- or trans-acting), and relevant literature, we ultimately identified four genes—CLDN1, IFIT5, RSAD2, and HCLS1—with six matched lncRNAs that may potentially regulate IAV replication in host cells. From these analyses, we generated a diagram representing the mutual mRNA and lncRNA interaction network (Figure 7). Within this network, RSAD2, CLDN1, HCLS1, and its matched lncRNAs play a role through cis-regulation, and IFIT5 and its matched lncRNAs play a role through trans regulation. These lncRNAs and their co-expressed mRNAs may reveal the molecular mechanism that affects IAV replication in MDCK cells and may also provide new genetic engineering targets. 

### 2.8. Validation of the Expression of 4 mRNAs with Their Matching Six lncRNAs at Different Time Points

Next, we compared the final screened four mRNAs and matched six lncRNAs at different infection time points (12 hpi, 16 hpi, 24 hpi). The results showed that among these four mRNAs and six lncRNAs, the most significant changes were observed at 16 hpi, which is consistent with previous experimental results (Figure 8).

### 2.9. Validation of the Expression of 4 mRNAs with Their Matching 6 lncRNAs in Different Virus Strains

After infecting MDCK cells with different virus strains (A (H1N1), BY, VSV) for 16 hpi, the expression of these four mRNAs and their matching six lncRNAs were consistent with the previous validation research results (Figure 9).

## 3. Discussion

We performed GO and KEGG enrichment analyses on the preliminarily screened DE mRNAs involved in the immune and virus responses and further studied the biological function and enrichment pathways of these DEGs. Functional enrichment analysis showed significant correlations with biological processes and functions, including metabolic processes, the interleukin-1 mediated signaling pathway, viral defense response, NF-κB binding, NF-κB signal transduction, the type I interferon signaling pathway, viral receptor activity, and catalytic activity. The NF-κB signaling pathway is an important cytokine signaling pathway that is activated by viral infections and a variety of cytokines and antigen receptors. Furthermore, it is involved in numerous biological processes, such as apoptosis, immune regulation, inflammatory reaction, and tumor formation [25]. It is also shown to inhibit the replication of many viruses in host cells, such as IAV, human immunodeficiency virus 1 (HIV-1), herpes simplex virus 1 (HSV-1), and other viruses [26]. In our study, MAP3K7, IRF2, and TLR2 were identified as key genes enriched in the NF-κB signaling pathway. The type I IFN signaling pathway is a classical antiviral and immunomodulatory signaling pathway. Canonical type I IFN signaling activates the JAK-STAT pathway, leading to transcription of ISGs, which exert antiviral effects [27]. For example, STAT1, STAT2, MAVS, and other related DE mRNAs were also identified in our omics data, but their functions and mechanisms are already well understood and do not require further detailed discussion here. The KEGG enrichment pathway analysis showed that our screened DEGs were significantly enriched in influenza A infection, the NF-κB signaling pathway, the TNF signaling pathway, HTLV-I infection, and other viral immune-related signaling pathways. Therefore, the above results indicate that MDCK host cells may respond to IAV infection via these biological processes and signaling pathways.

MAP3K7, also known as TAK1, has been reported to encode ubiquitously transforming growth factor β-activated kinase 1 and plays a critical role in both innate and adaptive immunity by regulating the inflammatory response, cell differentiation, cell survival, and apoptosis [28]. MAP3K7 is an effective immunosuppressive agent that exerts its antiviral effect via the TGF-β and MAPK signaling pathways [29]. TLR2 also plays a key role in recognizing bacterial components such as LPS, peptidoglycan, and lipoproteins. TLR2 is expressed in a variety of cells, including monocytes, dendritic cells (DCs), B cells, and T cells. TLR2 is a key gene in the TLR signaling pathway, the activation of which leads to the activation of MAPK and NF-κB signaling, thereby playing an antiviral role [30,31].

To further explore the role of DE mRNAs and lncRNAs after A(H1N1) infection, we constructed a targeted regulatory network of lncRNAs and mRNAs. This network showed that mRNAs can be targeted and regulated by multiple or single lncRNAs to play a potential role in regulating IAV replication. In addition, relevant studies have shown that one lncRNA can also target multiple mRNAs, thereby regulating the replication of some viruses. For example, a newly identified lncRNA IVRPIE regulates IFN-β 1 and transcription of several key ISGs, including IRF1, IFIT1, IFIT3, Mx1, ISG15, and IFI44L, to promote a host antiviral immune response [32]. Here, we used RT-qPCR and Western blot techniques to verify the trends in expression of these 32 lncRNAs and their 16 cis- or trans-regulated mRNAs and overall found consistent results with the predicted omics data. Therefore, for our remaining studies, we focused on the 32 virus response-related DE lncRNAs and their 16 target gene mRNAs that we screened with omics analysis and verified with RT-qPCR and Western blot. Ultimately, we focused on the four genes that were verified at both the molecular and protein levels, as well as the targeting relationship with their six matching lncRNAs. According to the omics data predictions, LTCONS_00140527 is directly bound to its target gene CLDN1 through cis. The position relationship between the two is that LTCONS_00140527 is in the just chain, while the target gene CLDN1 is in the antisense chain. The lncRNA and mRNA overlap by 2141 bp with a positive regulatory relationship between them. Based on relevant research, we speculate that LTCONS_00140527 is highly likely to directly bind to its target gene CLDN1 to play a role in the regulation of IAV replication, which is a critical discovery for subsequent in-depth research [33]. LTCONS_00085754 and LTCONS_00085755 are cis-acting lncRNAs that indirectly act on their target gene RSAD2. They are both located within 10 k bp upstream of RSAD2 and are positive regulators. LTCONS_00139462 and LTCONS_00139463 are also cis-acting lncRNAs that indirectly act on their target gene, HCLS1. Both lncRNAs are located within 20 k bp downstream of the HCLS1 and function as positive regulators. LTCONS_00127463 is a trans-acting lncRNA that indirectly acts on its target gene IFIT5 and is a positive regulator. Due to their distance from each other, their interaction can only be predicted based on binding energy calculation. These omics data predictions of lncRNA-mRNA correlations require further experimental studies, which will be explored in subsequent research.

Increasing evidence shows that these four genes are crucial for host resistance to multiple types of viral infections. IFIT5 is a factor in HCV entry, HCLS1 is a newly identified transcription regulator, and RSAD2 and IFIT5 are ISGs that have been shown to be critically important to resistance to viral infection, including influenza virus, HIV-1, and other viruses [34,35]. For example, RSAD2 performs its antiviral function by inhibiting the release of influenza virus through the JAK-STAT pathway mediated by type I/II IFN signaling [36,37]. IFIT5 inhibits the replication of many viruses in the host through the type I IFN signaling pathway. For example, it plays a key role in inhibiting the replication of the highly pathogenic porcine reproductive and respiratory syndrome virus (HP-PRRSV) that infects porcine pulmonary microvascular endothelial cells [38,39]. HAX-1 is an apoptosis inhibitor that is primarily located in the mitochondria. It interacts with the IAV PA protein and promotes replication of IAV H5N1 and avian influenza virus H9N2 in host cells [14,40]. CLDN1, a factor for HCV entry, is a key gene that promotes the replication of multiple viruses in host cells, including HCV and human parainfluenza virus type 2 (hPIV2) [35,41,42]. Furthermore, research has shown that the lncRNA LNC_000397 negatively regulates PRRSV replication by inducing the expression of ISGs, including RSAD2, MX1, and ISG15 [43].

The DE mRNAs and lncRNAs we identified and screened as potential factors affecting IAV replication function, as well as the potential regulatory lncRNA-mRNA interaction network, will require further analysis and verification to fully understand. Therefore, our follow-up experiments will use the results from this study as a starting point to further clarify the targeting mechanism between the lncRNAs and mRNAs and the interaction mechanism between host cells and viruses.

## 4. Methods

### 4.1. Cell Culture and Virus Infection

MDCK adherent cells (CCL-34, ATCC, Manassas, VA, USA) were stored in laboratory liquid nitrogen (−196 °C). DMEM (BGLM101.01, Lanzhou Bailing Biotechnology Co., Ltd., China) containing 10% fetal bovine serum (SA311.01, CellMax, Lanzhou, China) was used to culture the MDCK-P60 to a cell density of 80–90% at 5% CO_2_ and 37 °C. These conditions were also used for subculturing and carrying out subsequent experiments. The H1N1 subtype influenza virus X-275 strain (H1N1-X-275) and B/Phuket/3073/2013-like virus B (Yamagata/16/88 lineage) were obtained from Wuhan Institute of Biological Products Co., Ltd. (Wuhan, China). Vesicular stomatitis virus (VSV) was provided by the Key Laboratory of Biotechnology and Bioengineering of the National Ethnic Affairs Commission, Biomedical Research Center, Northwest Minzu University. MDCK cells were infected with the X-275, BY, and VSV strains (MOI = 0.01) or sham-infected with DMEM as a control. MDCK + H1N1 refers to the H1N1-X-275-infected MDCK cells 16 h post-infection (hpi), and MDCK refers to the DMEM-sham-infected MDCK cells (n = 3 per group). Labels for the two groups × replicates were as follows: MDCK + H1N1-1, MDCK + H1N1-2, MDCK + H1N1-3; and MDCK-1, MDCK-2, MDCK-3.

### 4.2. RNA Isolation, Reverse Transcription, and RT-qPCR

The total RNA of MDCK and MDCK + H1N1 cells was extracted using the TRIzol method (Invitrogen, Carlsbad, CA, USA), and the RNA concentration was determined using the MicroplateReader instrument (Thermo Fisher Scientific, Waltham, MA, USA) to determine the RNA concentration. Next, 1 μg of total RNA was reverse transcribed. Using the resulting cDNA as the template, we performed RT-qPCR using the fluorescent quantitative PCR reverse transcription kit and the All-in-One qPCR mix kit according to the manufacturer’s instructions. The RT-qPCR was performed in the Applied Biosystems 7500 Fluorescence Quantitative PCR Instrument (Thermo Fisher Scientific) in three independent experiments using the following reaction steps: initial denaturation at 95 °C for 30 s and 42 cycles of denaturation at 95 °C for 5 s, annealing at 60 °C for 35 s, and extension at 95 °C for 30 s. The qPCR results were normalized with the internal reference gene GAPDH and analyzed using the 2^−ΔΔCt^ method. The primers and target sequences used in this study are shown in (Appendix A).

### 4.3. Protein Expression Analysis Using Western Blot

The protein samples were collected with RIPA lysate and PMSF (ratio: 100:1), lysed on ice for 30 min, and centrifuged at 4 °C at 12,000 rpm for 10 min. The resulting supernatant was collected, from which the protein concentration was determined using the Bradford Assay and quantified uniformly. The proteins were separated using SDS-PAGE and transferred to a PVDF membrane using the wet transfer method. The membrane was then sealed with TBST solution containing 5% skim milk at room temperature for 2 h, treated with primary antibodies to a certain ratio: The primary antibodies were anti-RSAD2 (dilution ratio: 1:1000, 28674-1-AP, Wuhan Sanying, Wuhan, China), anti-IFIT5 (dilution ratio: 1:1000, 13378-1-AP, Wuhan Sanying), anti-HCLS1 (dilution ratio: 1:2000, 25003-1-AP, Wuhan Sanying), anti-CLDN1 (dilution ratio: 1:2000, 28674-1-AP, Wuhan Sanying), and anti-GAPDH (dilution ratio: 1:500); and the secondary antibodies were anti-rabbit and anti-mouse antibodies. Additionally, incubated at 4 °C overnight for 8–12 h. The membrane was subsequently treated with rabbit or mouse anti-IgG secondary antibody. Results were visualized using the ELC Detection Kit (PerkinElmer, Inc., Waltham, MA, USA) and Tanon 5500 Gel Imaging System (Tanon Science and Technology Co., Ltd., Shanghai, China). GAPDH was used as the loading control.

### 4.4. RNA Sequencing and Data Quality Analysis

After cell harvest and TRIzol extraction, the RNA was isolated and sequenced by companies that specialize in sequencing. Sample quality and species can influence the efficiency and stability of the experimental ribosome removal method, potentially leading to ribosome contamination that may affect subsequent analysis. Therefore, we first used the short reads comparison tool SOAPnuke v1.5.2 (https://github.com/BGI-flexlab/SOAPnuke) (accessed on 12 September 2023) [44] to compare the reads to the ribosome database, allowing up to five mismatches and using the following parameters: −l 15 −q 0.2 −n 0.05 −i. We then removed ribosome reads based on this comparison and used the retained data for subsequent analysis. Reads were then classified as dirty raw reads and removed if they met the following filter criteria: (1) reads with adapter; (2) reads containing > 10% N; and (3) low-quality reads (defined as alkali base with a quality value Q ≤ 10 accounting for >50% of the whole read). The data resulting from these three filtering steps were called clean reads and stored in the FASTQ format. See the help page for the description of the FASTQ format. This sequence file may be directly used for publishing, public database submission, and subsequent analysis.

### 4.5. RNA-Seq Differential Gene Expression Analysis

Differential analysis was used to compare the relative gene expression between the control (MDCK) and Target (MDCK + H1N1) samples. Genes with insignificant expression changes were excluded, and genes with significant differences were retained. We used the difference analysis method PostionDis to analyze the differences between samples using the following filter conditions for significantly differentially expressed genes: Fold Change (FC) ≥ 2.00 and false discovery rate (FDR) ≤ 0.001.

### 4.6. lncRNA Target Gene Prediction

lncRNAs regulate target genes in either a cis- or trans-acting manner. The basic principle of cis-target gene prediction is that the function of a cis-acting lncRNA is related to the protein-coding genes adjacent to its coordinates. Therefore, the mRNA adjacent to the predicted cis-acting lncRNA was selected as its target gene. Since trans regulation does not require close proximity to the target gene, we predicted trans-acting lncRNAs by calculating the binding energy. If there was an overlap between the lncRNA and its target gene, we classified the level of overlap in detail to clarify the detailed mechanism of cis-regulation [24,45]. To analyze the target genes, we calculated two correlation coefficients between the lncRNAs and mRNAs, namely, Spearman and Pearson, using the following threshold parameters: Spearman required_Cor ≥ 0.6 and Pearson_cor ≥ 0.6. To analyze the cis and trans regulation, we defined lncRNAs within 10 k bp upstream or 20 k bp downstream of its mRNA as cis-acting. Furthermore, for lncRNAs and mRNAs beyond this range, we used RNAplex to analyze the lncRNA-mRNA binding energy and defined lncRNAs with binding energy < –30 as trans-acting.

### 4.7. Gene Ontology and KEGG Enrichment Analysis of the RNA-Seq Data

We chose the Gene Ontology (GO) and KEGG (www.kegg.jp/kegg/kegg1.html) (accessed on 12 September 2023) databases for our gene function analysis of our RNA-seq results [46,47,48], using either a GO term or a pathway as a functional module. Our candidate genes of interest included both significantly differentially expressed genes (DEGs) between infected and uninfected cells as well as target genes of differentially expressed lncRNAs. We conducted an enrichment analysis to identify the functional modules with the highest concentration of candidate genes. Significant enrichment of candidate genes in a functional module indicates that the proportion of candidate genes in that functional module was significantly higher than the background, which is the reference that includes the overall proportion of candidate genes. In order to quantitatively evaluate whether candidate genes of a functional module were enriched, we used a hypergeometric model to calculate the degree of enrichment of candidate genes in each functional module. Our analysis resulted in a *p*-value for each functional module, such that smaller *p*-values indicate more abundant candidate genes in this functional module. *p*-values were FDR-corrected, and functional modules were found to have significant enrichment with FDR ≤ 0.01.

### 4.8. Coding Ability Prediction

To distinguish between mRNA and lncRNA for each new candidate transcript, we predicted its encoding ability using three forecasting software platforms with default parameters—CPC v0.9-r2 (http://CPC.cbi.pku.edu.cn) (accessed on 12 September 2023) [49], txCdsPredict (http://hgdownload.soe.ucsc.edu/admin/jksrc.zip) (accessed on 12 September 2023), and CNCI (https://github.com/www-bioinfo-org/CNCI) (accessed on 12 September 2023) [50]—and the protein database Pfam (http://pfam.xfam.org/) (accessed on 12 September 2023) [51] with default parameters. All three prediction software platforms score the coding ability of transcripts and then set a scoring threshold to distinguish lncRNA and mRNA. Transcripts with matching Pfam results were considered to be mRNA; otherwise, they were identified as lncRNA. At least three of the four methods were consistent, which allowed us to confirm the identity of a transcript as mRNA or lncRNA.

The scoring thresholds of the three software platforms were as follows: CPC_Threshold = 0, transcripts > 0 are mRNA and transcripts < 0 are lncRNA; CNCI_Threshold = 0, transcripts > 0 are mRNA and transcripts < 0 are lncRNA; and txCdsPredict_Threshold = 500, transcripts > 500 are mRNA, and transcripts < 500 are lncRNA.

### 4.9. Statistical Analysis

The statistical analysis was conducted using GraphPad Prism Software 8.0 (GraphPad Prism Software, State of California, USA). All data in the study were expressed as mean ± standard deviation from at least three independent experiments. Statistical comparisons were made using an unpaired Student’s *t*-test (for comparisons between two groups) or a one-way ANOVA (for comparisons among multiple groups). *p* < 0.05 was considered statistically significant.

## 5. Conclusions

In this study, we speculate that the 18 screened DE mRNAs and 32 DE lncRNAs may affect IAV replication in MDCK cells through the biological processes and signaling pathways we identified. According to the omics analysis results, a potential lncRNA-mRNA interaction network related to the regulation of IAV replication was predicted and thoroughly analyzed. This network was composed of four target DEGS—CLDN1, IFIT5, RSAD2, and HCLS1—and their six matched lncRNAs LTCONS_00140527, LTCONS_00127643, LTCONS_00085754, LTCONS_00085755, LTCONS_00139462, and LTCONS_00139463. The results of this study will contribute to a more comprehensive understanding of the molecular mechanism of host non-coding RNA-mediated regulation of influenza virus replication and will also provide potential methods for the screening and identification of target genes for the development of genetically engineered cell lines capable of high-yield artificial vaccine production.

## Figures and Tables

**Figure 1 vaccines-11-01593-f001:**
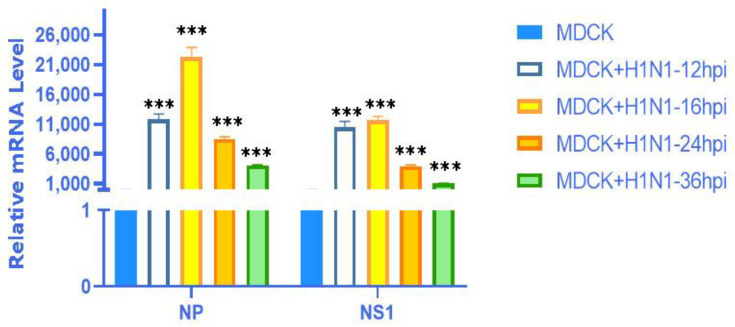
RT-qPCR was used to detect the expression changes of NP and NS1 genes of the A (H1N1) virus at different infection time points (12 hpi, 16 hpi, 24 hpi, 36 hpi) on MDCK + H1N1 cells. The data are expressed as x ± s (n = 3). ***: *p* < 0.001.

**Figure 2 vaccines-11-01593-f002:**
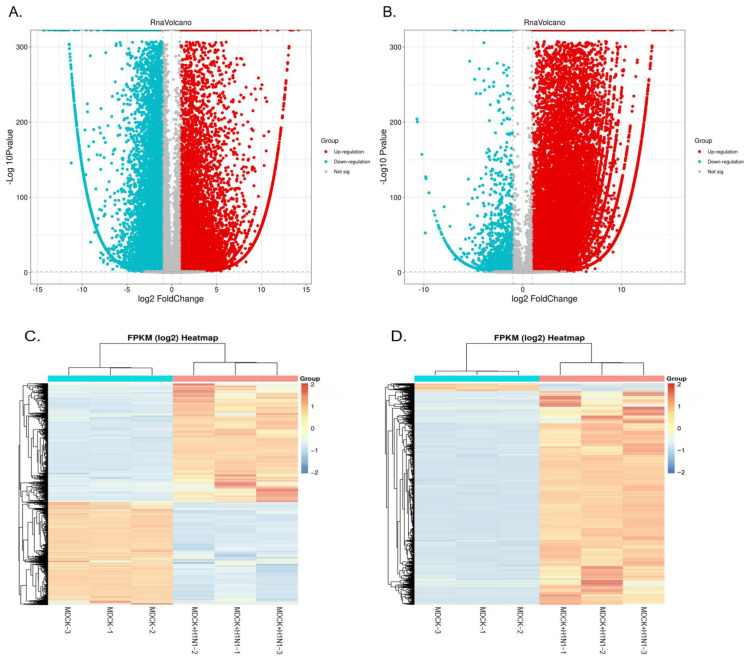
DE mRNAs and lncRNAs in MDCK cells infected with A(H1N1). MDCK cells were infected with A(H1N1) (X-275 strain; MOI = 0.01) or sham-infected with DMEM and harvested 16 hpi for analysis. Volcano plots of DE (**A**) mRNAs and (**B**) lncRNAs in MDCK + H1N1 cells relative to uninfected MDCK cells. (**C**) Heat map of DE mRNAs in MDCK + H1N1 cells relative to uninfected MDCK cells; each column represents a different sample, and each row represents a different mRNA. (**D**) Heat map of DE lncRNAs in MDCK + H1N1 cells relative to uninfected MDCK cells; each column represents a different sample, and each row represents a different lncRNA. Transcripts were considered significantly different using the following criteria: |log2FC| > 1 and *p* < 0.05. Abbreviations: DE, differentially expressed; IAV, influenza A virus; hpi, hours post-infection; MDCK + H1N1, infected MDCK cells; FC, fold change.

**Figure 3 vaccines-11-01593-f003:**
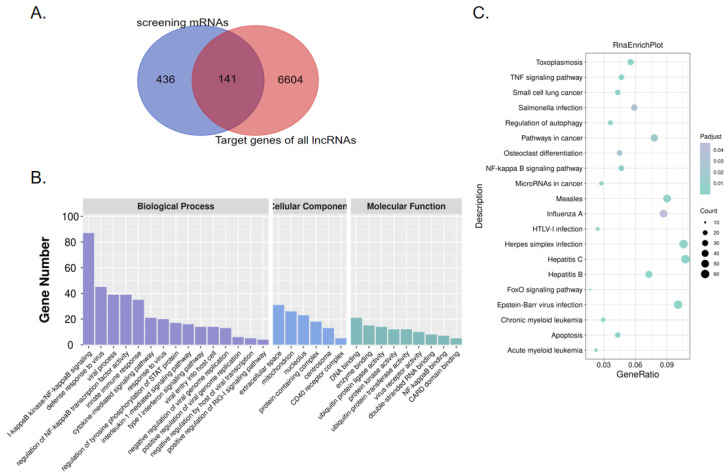
Screening of DE mRNAs related to virus response based on functional enrichment analysis. (**A**) Venn diagram of intersection analysis between virus-response related DE mRNAs in MDCK cells and predicted matched lncRNAs. Blue represents the 577 preliminarily screened virus response-related DE mRNAs; pink represents the predicted target gene (mRNA) of the total DE lncRNAs, and dark red (middle) represents DE mRNAs related to virus response and matching predicted DE lncRNAs. (**B**) GO function enrichment analysis of the 577 selected DE mRNAs. (**C**) Bubble diagram of the KEGG pathway enrichment analysis of the 577 preliminarily screened DE mRNAs, showing the top 20 KEGG pathways with the most significant enrichment. Abbreviations: DE, differentially expressed; GO, Gene Ontology.

**Figure 4 vaccines-11-01593-f004:**
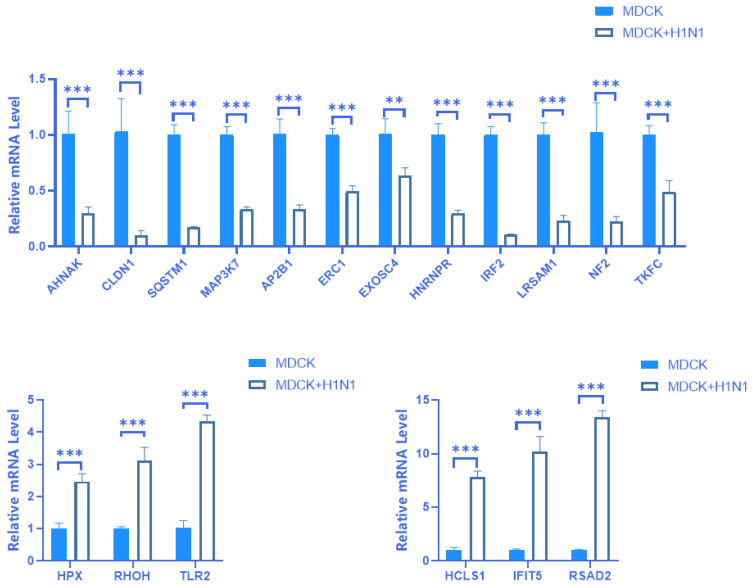
Verification of DE mRNA expression using RT-qPCR in A(H1N1)-infected MDCK cells. RT-qPCR was performed on 18 screened DEGs identified from RNA-seq data in MDCK cells infected with A(H1N1) 16 hpi relative to uninfected MDCK cells. Blue represents MDCK cells, and white represents MDCK + H1N1 cells. Relative expression was determined using the 2^−ΔΔCt^ method with GAPDH as the internal reference gene. The data are expressed as mean ± standard deviation (n = 3). Statistical significance was determined using GraphPad Prism 8.0. *** *p* < 0.001. Abbreviations: DE, differentially expressed; IAV, influenza A virus; DEGs, differentially expressed genes; hpi, hours post-infection; MDCK + H1N1, infected MDCK cells.

**Figure 5 vaccines-11-01593-f005:**
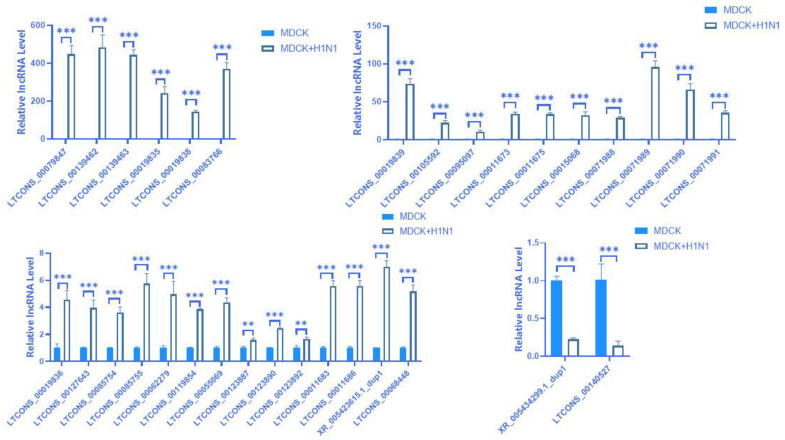
Verification of DE lncRNA expression using RT-qPCR in A(H1N1)-infected MDCK cells. RT-qPCR was performed on 37 screened DE lncRNAs identified from RNA-seq data in MDCK cells infected with A(H1N1) 16 hpi relative to uninfected MDCK cells. Blue represents MDCK cells, and white represents MDCK + H1N1 cells. Relative expression was determined using the 2^−ΔΔCt^ method with GAPDH as the internal reference gene. The data are expressed as mean ± standard deviation (n = 3). Statistical significance was determined using GraphPad Prism 8.0. ** *p* < 0.01; *** *p* < 0.001. Abbreviations: DE, differentially expressed; IAV, influenza A virus; hpi, hours post-infection; MDCK + H1N1, infected MDCK cells.

**Figure 6 vaccines-11-01593-f006:**
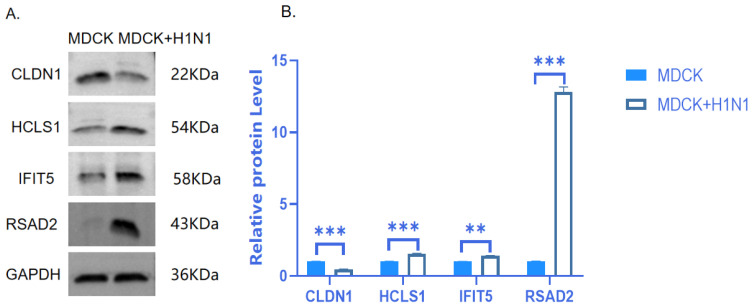
Verification of CLDN1, HCLS1, IFIT5, and RSAD2 differential protein expression in A(H1N1)-infected MDCK cells. Western blot was performed on four screened DEGs in MDCK cells infected with A(H1N1) 16 hpi relative to uninfected MDCK cells. (**A**) SDS-PAGE gel and (**B**) quantitative protein expression levels are shown. GAPDH was used as the reference protein. Blue represents MDCK cells, and white represents MDCK + H1N1 cells. The data are expressed as mean ± standard deviation (n = 3). Statistical significance was determined using GraphPad Prism 8.0. ** *p* < 0.01; *** *p* < 0.001. Abbreviations: IAV, influenza A virus; DEGs, differentially expressed genes; hpi, hours post-infection; MDCK + H1N1, infected MDCK cells.

**Figure 7 vaccines-11-01593-f007:**
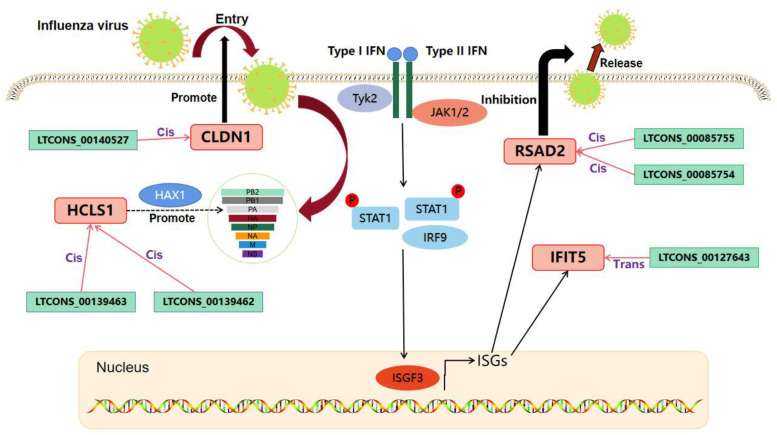
Interaction network of genes and co-expressed lncRNAs involved in IAV replication regulation in MDCK cells. Pink represents genes, and green represents lncRNA. RSAD2, CLDN1, IFIT5, and HCLS1 exhibit consistent expression patterns with their predicted matching lncRNAs, indicating that they may have a positive regulatory target gene relationship with their matching lncRNAs. RSAD2 and IFIT5 may inhibit IAV replication in MDCK cells through the type I/II interferon signaling pathway, in which RSAD2 inhibits the release of IAV from the plasma membrane of host cells. CLDN1, a key factor involved in HCV entry, may also assist in IAV entry into MDCK cells. HAX-1 interacts with the PA subunit of IAV, which may promote the replication of IAV in MDCK cells. Abbreviation: IAV, influenza A virus; HCV, hepatitis C virus; HAX-1, HCLS1 interacting protein X-1.

**Figure 8 vaccines-11-01593-f008:**
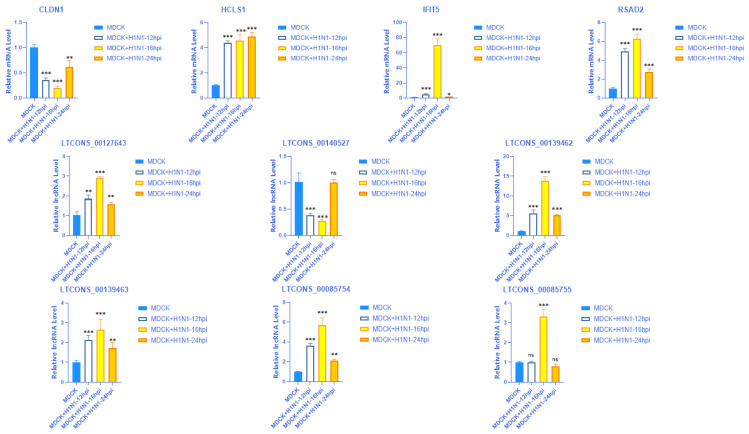
The expression levels of four mRNA (CLDN1, HCLS1, IFIT5, RSAD2) and their matching six lncRNAs (LTCONS-00127643, LTCONS-00140527, LTCONS-00139462, LTCONS-00139463, LTCONS-00085754, LTCONS-00085755) were detected using RT qPCR at different infection times (12 hpi, 16 hpi, 24 hpi) in MDCK + H1N1 cells. The data are expressed as x ± s (n = 3). *: *p* < 0.05; **: *p* < 0.01; ***: *p* < 0.001 (b).

**Figure 9 vaccines-11-01593-f009:**
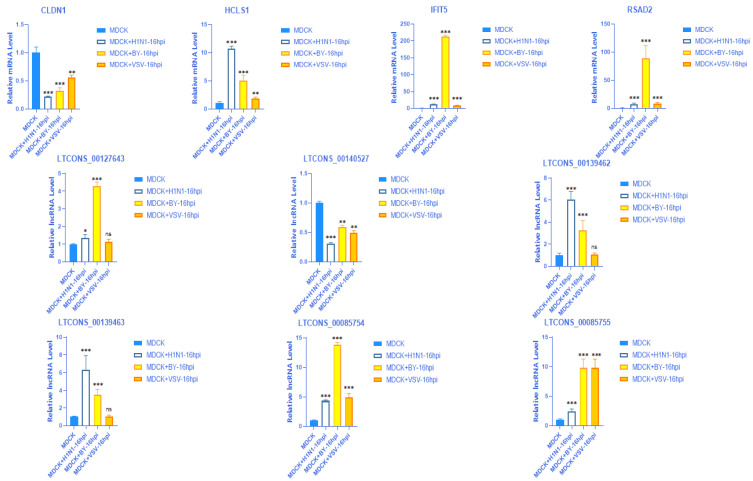
The expression levels of four mRNA (CLDN1, HCLS1, IFIT5, RSAD2) and their matching six lncRNAs (LTCONS-00127643, LTCONS-00140527, LTCONS-00139462, LTCONS-00139463, LTCONS-00085754, LTCONS-00085755) were detected using RT qPCR when infected with different virus strains (A (H1N1), BY, VSV) on MDCK cells at 16 hpi. * *p* < 0.05; ** *p* < 0.01; *** *p* < 0.001; ns (no significance).BY: Influenza B/Phuket/3073/2013-like virus B (Yamagata/16/88 lineage); VSV: Vesicular stomatitis virus.

**Table 1 vaccines-11-01593-t001:** DE mRNA targeting virus response and its matched DE lncRNA.

lncRNA	log2(lncRNA) Ratio (MDCK + H1N1/MDCK)	*p*-Value (lncRNA)	lncRNA_Class	Target Gene	Gene ID	log2(mRNA) Ratio (MDCK + H1N1/MDCK)	*p*-Value (mRNA)
LTCONS_00079847	8.49	7.12 × 10^−53^	cis_mRNA_up10k	TLR2	XM_038687174.1_dup1	8.04	2.45 × 10^−21^
LTCONS_00139462	11.49	2.62 × 10^−134^	cis_mRNA_dw20k	HCLS1	XM_038445109.1_dup1	7.33	1.79 × 10^−129^
LTCONS_00139463	8.12	8.97 × 10^−166^	cis_mRNA_dw20k				
LTCONS_00019835	8.58	1.11 × 10^−28^	cis_mRNA_dw20k	RHOH	XM_038661947.1_dup1	6.56	1.91 × 10^−111^
LTCONS_00019836	7.13	6.97 × 10^−13^	cis_mRNA_dw20k				
LTCONS_00019838	6.83	1.09 × 10^−168^	cis_mRNA_up10k				
LTCONS_00019839	6.62	1.04 × 10^−121^	cis_mRNA_up10k				
LTCONS_00127643	2.43	1.83 × 10^−90^	tran	IFIT5	XM_038662420.1_dup1	5.95	1.49 × 10^−22^
LTCONS_00105592	4.48	1.41 × 10^−26^	cis_mRNA_up10k	HPX	XM_038430363.1_dup1	5.84	7.47 × 10^−31^
LTCONS_00085754	1.58	1.17 × 10^−6^	cis_mRNA_up10k	RSAD2	XM_038690652.1_dup1	4.99	0
LTCONS_00085755	3.74	1.21 × 10^−76^	cis_mRNA_up10k				
LTCONS_00062279	3.04	0	cis_mRNA_dw20k	SQSTM1 (p62)	XM_038681126.1_dup1	−2.23	3.95 × 10^−114^
XR_005434299.1_dup1	−1.31	1.30 × 10^−204^	cis_mRNA_overlap				
LTCONS_00119854	3.34	1.42 × 10^−25^	cis_mRNA_dw20k	NF2	XM_038437180.1_dup1	−2.25	0
LTCONS_00055069	3.81	1.70 × 10^−103^	cis_mRNA_up10k	AP2B1	XM_038677309.1_dup1	−2.74	0
LTCONS_00140527	−2.69	1.43 × 10^−9^	cis_mRNA_overlap	CLDN1	XM_038445869.1_dup1	−2.77	0
LTCONS_00083766	7.49	9.19 × 10^−16^	cis_mRNA_dw20k	IRF2	XM_038690244.1_dup1	−2.88	1.63 × 10^−255^
LTCONS_00123887	1.14	2.09 × 10^−10^	cis_mRNA_overlap	ERC1	XM_038439411.1_dup1	−2.99	9.22 × 10^−182^
LTCONS_00123890	1.62	3.86 × 10^−9^	cis_mRNA_overlap				
LTCONS_00123892	1.39	5.04 × 10^−8^	cis_mRNA_overlap				
LTCONS_00095097	5.69	6.72 × 10^−68^	cis_mRNA_overlap	AHNAK	XM_038425334.1_dup1	−3.55	0
LTCONS_00011673	7.08	3.96 × 10^−57^	cis_mRNA_up10k	HNRNPR	XM_038659953.1_dup1	−3.85	0
LTCONS_00011675	5.39	1.32 × 10^−8^	cis_mRNA_up10k				
LTCONS_00011683	2.29	2.70 × 10^−193^	cis_mRNA_dw20k				
LTCONS_00011686	4.43	0	cis_mRNA_dw20k				
LTCONS_00015068	4.72	2.76 × 10^−53^	cis_mRNA_up10k				
XR_005423615.1_dup1	3.10	0	cis_mRNA_dw20k				
LTCONS_00068448	2.81	1.53 × 10^−12^	cis_mRNA_up10k	MAP3K7 (MKK7)	XM_038684179.1_dup1	−3.93	0
LTCONS_00071988	5.98	1.55 × 10^−97^	cis_mRNA_dw20k	EXOSC4	XM_038685312.1_dup1	−10.33	2.09 × 10^−168^
LTCONS_00071989	7.25	1.30 × 10^−74^	cis_mRNA_dw20k				
LTCONS_00071990	4.22	1.56 × 10^−51^	cis_mRNA_dw20k				
LTCONS_00071991	5.38	8.11 × 10^−72^	cis_mRNA_dw20k				

Note: log2Ratio (MDCK + H1N1/MDCK): The multiple values of the difference were rounded uniformly, keeping two decimal places; up: lncRNA is located upstream of its target gene; dw: lncRNA is located downstream of its target gene. Abbreviations: DE, differentially expressed; MDCK + H1N1, MDCK cells infected with influenza A virus H1N1 16 h post-infection; MDCK, uninfected MDCK cells.

## Data Availability

All data generated or analyzed during this study are included in this article and its additional files. All RNA-seq data analyzed during this study has been deposited in the SRA database (https://www.ncbi.nlm.nih.gov/sra) (accessed on 12 September 2023) under accession numbers: SRR22541678, SRR22541679, SRR22541680, SRR22541681, SRR22541682, SRR22541683.

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
