# Peer review of "Transcriptional Analysis of lncRNA and Target Genes Induced by Influenza A Virus Infection in MDCK Cells"

_vaccines, 2023, doi:10.3390/vaccines11101593_

Round 1

Reviewer 1 Report

The author investigated the DEGs mRNA by comparing MDCK+H1N1-infected cells with uninfected MDCK cells and identified matching predicted DE lncRNAs. The rationale for the study is evident as it aims to explore differentially expressed (DE) lncRNAs and mRNAs and assess the relationship between significant lncRNAs and their target genes. The study's introduction and the results are clearly described and presented, and the discussion is appropriate. The abstract accurately describes the results and conclusions. Some comments have been provided as follows.

1.     There were 141 virus response-related DE mRNAs matching predicted DE lncRNAs. To help the reader, please provide the rationale for selecting 18 DEGs mRNA for subsequent analysis. In addition, please clarify why only four mRNAs were chosen to observe the role of molecular mechanism while others were omitted.

2.     Could the author explain or provide a brief description of how four specific mRNAs are involved in regulating the IAV replication in lines 276-280?

 3.     In statistical analysis, the author used Pearson correlation analysis to identify the co-expressed lncRNAs and mRNA with the threshold parameters Spearman Cor ≥ 0.6 and Pearson Cor ≥ 0.6. Is this threshold reliable?

 4.     The results require wet experiments to confirm the interaction between predicted lncRNAs and their target genes in regulating IAV replication in MDCK cells, such as knocking down lncRNA and observing their matched target gene expression. Please add this point as a limitation or suggest it as a direction for further study.

Author Response

  1. There were 141 virus response-related DE mRNAs matching predicted DE lncRNAs. To help the reader, please provide the rationale for selecting 18 DEGs mRNA for subsequent analysis. In addition, please clarify why only four mRNAs were chosen to observe the role of molecular mechanism while others were omitted.

Response: Thank you for your comments. We preliminarily screened 141 differentially expressed mRNA based on transcriptome analysis results. On this basis, we selected the top 20 genes with the highest differences both up-regulation and down-regulation based on the multiple and significance of expression differences, totaling 40 genes. Considering the possibility of some randomness and false positive results in high-throughput transcriptome analysis results.We used Realtime PCR to perform three verifications on samples of different cell generations, and  screened a total of 18 mRNA with uniform expression differences and consistent transcriptome analysis results. Based on the maximum differential multiple and existing research reports, we preliminarily selected four genes for further protein level validation in the screened differential mRNA. However, our study did not overlook the other genes among these 18 mRNA genes. We will conduct individual validation and research on the expression of these genes, their regulatory function in influenza virus response, and their regulatory relationship with predicted lncRNA in subsequent studies. And it has been modified, supplemented, and highlighted in yellow in the Results(2.4. Verification of mRNA expression of 18 screened DEGs) section of the main text.

  1. Could the author explain or provide a brief description of how four specific mRNAs are involved in regulating the IAV replication in lines 276-280?

Response: Thank you for your comments. lncRNAs can combine with DNA, RNA, and proteins to regulate gene expression at multiple levels, including epigenetic, transcriptional, and post-transcriptional regulation. Studies have shown that lncRNAs play a key regulatory role in the fight between host cells and viruses. Recently, they have been shown to interact with a variety of PRRs, and both directly or indirectly act on their target genes or related signaling pathways to regulate IAV replication. For example, lnc-Lsm3b inhibits the initiation of antiviral signal transduction downstream of RIG-I by competing with viral RNA for the binding of RIG-I monomers, thus promoting the replication of A(H1N1). Additionally, lnc-MxA is an ISG that forms an RNA-DNA triplet, which inhibits IFN-β and subsequent activation of its related signaling pathway, thereby promoting IAV replication. (This paragraph has been reflected in lines 81 to 91 of the main text). RSAD2 and IFIT5 may inhibit IAV replication in MDCK cells through the type I/II interferon signaling pathway, in which RSAD2 inhibits the release of IAV from the plasma membrane of host cells. CLDN1, a key factor involved in HCV entry, may also assist IAV entry into MDCK cells. HAX-1 interacts with the PA subunit of IAV, which may promote the replication of IAV in MDCK cells. (This paragraph has been reflected in lines 288 to 292 of the main text). The function of LncRNA is mainly achieved by cis or trans acting on the target gene. Cis acting on the target gene prediction principle consider the function of lncRNA is related to the protein coding gene near its coordinates, so mRNA adjacent to lncRNA are selected as target genes. And trans regulation is not dependent on the location relationship, we predict it by combining energy method. (This paragraph has been reflected in lines 488 to 495 of the main text).

  1. In statistical analysis, the author used Pearson correlation analysis to identify the co-expressed lncRNAs and mRNA with the threshold parameters Spearman Cor ≥ 0.6 and Pearson Cor ≥ 0.6. Is this threshold reliable?

Response: Thank you for your comments. This threshold is reliable, multiple articles have adopted this threshold parameter in their analysis methods 1-3.

  1. The results require wet experiments to confirm the interaction between predicted lncRNAs and their target genes in regulating IAV replication in MDCK cells, such as knocking down lncRNA and observing their matched target gene expression. Please add this point as a limitation or suggest it as a direction for further study.

Response: Thank you. Your suggestion is very professional. The regulatory relationship between the screened lncRNAs and mRNA, as well as the molecular mechanisms that affect the replication of influenza viruses in cells, will be the focus of future research. We mainly consider knocking out or overexpressing lncRNA to detect its impact on virus replication. In fact, this work is ongoing, but due to the huge and cumbersome workload, we will report on lncRNAs that can significantly affect virus replication in future papers. Thank you again for your suggestion.

  1. Wang, Q. C.; Wang, Z. Y.;  Xu, Q.;  Chen, X. L.; Shi, R. Z., lncRNA expression profiles and associated ceRNA network analyses in epicardial adipose tissue of patients with coronary artery disease. Sci Rep 2021, 11 (1), 1567.
  2. Kornienko, A. E.; Guenzl, P. M.;  Barlow, D. P.; Pauler, F. M., Gene regulation by the act of long non-coding RNA transcription. BMC biology 2013, 11, 59.
  3. Knauss, J. L.; Sun, T., Regulatory mechanisms of long noncoding RNAs in vertebrate central nervous system development and function. Neuroscience 2013, 235, 200-14.

Reviewer 2 Report

The article is serious and provides clues about treatment with vaccines to prevent influenza. The results of this study will contribute to a more comprehensive understanding of the molecular mechanism of host cell non-coding RNA-mediated regulation of influenza virus replication. These results may also identify methods for screening target genes in the development of genetically engineered cell lines capable of high yield artificial vaccine production. 

The references are current and therefore, the article is based on recent publications. As a negative aspect, I would say that the format of the references does not adapt to the general one, with greater spacing than what is observed in the text. I would therefore ask that the presentation be made uniform.

Author Response

The article is serious and provides clues about treatment with vaccines to prevent influenza. The results of this study will contribute to a more comprehensive understanding of the molecular mechanism of host cell non-coding RNA-mediated regulation of influenza virus replication. These results may also identify methods for screening target genes in the development of genetically engineered cell lines capable of high yield artificial vaccine production. 

The references are current and therefore, the article is based on recent publications. As a negative aspect, I would say that the format of the references does not adapt to the general one, with greater spacing than what is observed in the text. I would therefore ask that the presentation be made uniform.

Response: Thank you for your comments. The format of the references has been modified in the main text.

Reviewer 3 Report

Influenza A virus infection is an important medical and social problem, so a better understanding of the mechanisms of pathogenesis of viral infection will improve approaches to its treatment. This article analyzes gene and lncRNA expression in cell culture during influenza A virus infection, which is of research practical interest. The study design accomplishes the objectives stated by the authors.

Comments

1. It is recommended to add a table with protein names and functions for the 18 screened DEGs.

2. It is also recommended that the prospects for future research be discussed in light of the findings, given the conclusions that the results may identify methods for screening target genes in the development of genetically engineered cell lines capable of high yield artificial vaccine production.

Author Response

Influenza A virus infection is an important medical and social problem, so a better understanding of the mechanisms of pathogenesis of viral infection will improve approaches to its treatment. This article analyzes gene and lncRNA expression in cell culture during influenza A virus infection, which is of research practical interest. The study design accomplishes the objectives stated by the authors.

Comments

  1. It is recommended to add a table with protein names and functions for the 18 screened DEGs.

Response: Thank you for your comments. We have added a table (Additional file 7 Table S7) to the supplementary data, indicating the protein names and basic functional descriptions of the differentially expressed genes.

  1. It is also recommended that the prospects for future research be discussed in light of the findings, given the conclusions that the results may identify methods for screening target genes in the development of genetically engineered cell lines capable of high yield artificial vaccine production.

Response: Thank you for your comments. In the era of high-throughput biology (from sequencing to screening) it is hoped that targets can be identified, validated and translated into clinical candidates in ever decreasing periods of time. However, it should be acknowledged that it often takes many years to understand fully the biology of a particular target and to select the best utility of this in particular diseases. A key example in the field of inflammation is TNF-α, which has efficacy in a variety of autoimmune diseases through the use of anti-TNF-α biologics4. Antiviral therapeutics can be directed at the host or at the virus itself, and broad-spectrum activity if the target is used by multiple viruses5. Therefore, lncRNA, as a new target for regulating influenza virus proliferation in host cells, provides a strong theoretical basis for screening target genes for establishing vaccine high yield cell lines.

  1. DL, S., What makes a good anti-inflammatory drug target? Drug discovery today 2006, 11 (5-6), 210-9.
  2. von Delft, A.; Hall, M. D.;  Kwong, A. D.;  Purcell, L. A.;  Saikatendu, K. S.;  Schmitz, U.;  Tallarico, J. A.; Lee, A. A., Accelerating antiviral drug discovery: lessons from COVID-19. Nat Rev Drug Discov 2023, 22 (7), 585-603.

Reviewer 4 Report

The authors conducted a comprehensive analysis of lncRNAs involved in influenza virus proliferation and their target RNAs using the MDCK cell used for influenza vaccine production. As a result, six lncRNAs and four target RNAs were identified. This is a very interesting study for me and probably other readers. In the future, it is expected that regulating the expression of these factors will lead to the establishment of cell lines with improved virus proliferation. When considering the goal of this research to improve the efficiency of vaccine production, I think it is important to ensure the safety of the vaccine, including the genetic stability of the virus.

[Major point]

Nothing

[Minor point]

This is a matter of editing, but the format of the references should be arranged.

Author Response

The authors conducted a comprehensive analysis of lncRNAs involved in influenza virus proliferation and their target RNAs using the MDCK cell used for influenza vaccine production. As a result, six lncRNAs and four target RNAs were identified. This is a very interesting study for me and probably other readers. In the future, it is expected that regulating the expression of these factors will lead to the establishment of cell lines with improved virus proliferation. When considering the goal of this research to improve the efficiency of vaccine production, I think it is important to ensure the safety of the vaccine, including the genetic stability of the virus.

Response: Thank you for your comments. Indeed, as a development of cell lines for vaccine production, the impact of changes in target genes on the biological safety of cells themselves is a very noteworthy issue. At present, research has shown whether screening genes are related to cell tumorigenesis. In the next step of research, we will not only focus on screening the effects of lncRNA and mRNA on virus replication, but also focus on the effects on cell proliferation, migration and invasion, as well as the tumorigenic phenotype of nude mice.

[Major point]

Nothing.

[Minor point]

This is a matter of editing, but the format of the references should be arranged.

Response: Thank you for your comments. The format of the references has been modified in the main text.

Reviewer 5 Report

Dear authors

I hope all of you are always fine. Regarding the revision of the manuscript No. vaccines-2618622, titled “Transcriptional analysis of lncRNA and target genes induced by influenza A virus infection in MDCK cells”. Really, it is an interesting research; however, some comments should be replied.

Comments:

1-    Why did you select only 4 protein genes (RSAD2, HCLS1, IFIT5 and CLDN1) for Western blot verification?

2-    Why the results of IFIT5, LTCONS-00127643, LTCONS-00139462 in figure 9 are significantly lower in MDCK+H1N1-16hpi than the same time in figure 8. Please discuss.

3-    Why the results of LTCONS-00139463 in figure 9 are significantly higher in MDCK+H1N1-16hpi than the same time in figure 8. Please discuss.

Moderate English language editing is required. e.g. line 43, contain should be replaced by control.

Author Response

Comments:

1.Why did you select only 4 protein genes (RSAD2, HCLS1, IFIT5 and CLDN1) for Western blot verification?

Response: Thank you for your comments. We preliminarily screened 141 differentially expressed mRNA based on transcriptome analysis results. On this basis, we selected the top 20 genes with the highest differences both up-regulation and down-regulation based on the multiple and significance of expression differences, totaling 40 genes. Considering the possibility of some randomness and false positive results in high-throughput transcriptome analysis results. We used Realtime PCR to perform three verifications on samples of different cell generations, and screened a total of 18 mRNA with uniform expression differences and consistent transcriptome analysis results. Based on the maximum differential multiple and existing research reports, we preliminarily selected four genes for further protein level validation in the screened differential mRNA. However, our study did not overlook the other genes among these 18 mRNA genes. We will conduct individual validation and research on the expression of these genes, their regulatory function in influenza virus response, and their regulatory relationship with predicted lncRNA in subsequent studies. And it has been modified, supplemented, and highlighted in yellow in the Results(2.4. Verification of mRNA expression of 18 screened DEGs) section of the main text.

2.Why the results of IFIT5, LTCONS-00127643, LTCONS-00139462 in figure 9 are significantly lower in MDCK+H1N1-16hpi than the same time in figure 8. Please discuss.

Response: Thank you for your comments. Due to the different cell generations used in Figures 8 and 9, there may be differences in multiples, but the overall trend is consistent and in line with the expected trend.

3.Why the results of LTCONS-00139463 in figure 9 are significantly higher in MDCK+H1N1-16hpi than the same time in figure 8. Please discuss.

 Response: Thank you for your comments. Due to the different cell generations used in Figures 8 and 9, there may be differences in multiples, but the overall trend is consistent and in line with the expected trend.

Comments on the Quality of English Language:

Moderate English language editing is required. e.g. line 43, contain should be replaced by control.

Response: Thank you for your comments. Corrected and highlighted in yellow in the main text.